# Physicochemical Properties of Gold Nanoparticles for Skin Care Creams

**DOI:** 10.3390/ma16083011

**Published:** 2023-04-11

**Authors:** Peter Majerič, Zorana Jović, Tilen Švarc, Žiga Jelen, Andrej Horvat, Djuro Koruga, Rebeka Rudolf

**Affiliations:** 1Faculty of Mechanical Engineering, University of Maribor, 2000 Maribor, Slovenia; peter.majeric@um.si (P.M.); tilen.svarc1@um.si (T.Š.); z.jelen@um.si (Ž.J.); 2Zlatarna Celje d.o.o., 3000 Celje, Slovenia; 3TFT Nano Center, 11050 Belgrade, Serbia; zorana.jovic@tftnanocenter.rs (Z.J.); djuro.koruga@gmail.com (D.K.); 4Zepter-Slovenica d.o.o., 2380 Slovenj Gradec, Slovenia; ahorvat@zepter.si; 5Nano Lab, Faculty of Mechanical Engineering, University of Belgrade, 11000 Belgrade, Serbia

**Keywords:** gold nanoparticles, ultrasonic spray pyrolysis, freeze drying, characterization, creams

## Abstract

Gold nanoparticles (AuNPs) have now been used in skin care creams for several years, with marketed anti-aging, moisturizing, and regenerative properties. Information on the harmful effects of these nanoparticles is lacking, a concern for the use of AuNPs as cosmetic ingredients. Testing AuNPs without the medium of a cosmetic product is a typical method for obtaining this information, which is mainly dependent on their size, shape, surface charge, and dose. As these properties depend on the surrounding medium, nanoparticles should be characterized in a skin cream without extraction from the cream’s complex medium as it may alter their physicochemical properties. The current study compares the sizes, morphology, and surface changes of produced dried AuNPs with a polyvinylpyrrolidone (PVP) stabilizer and AuNPs embedded in a cosmetic cream using a variety of characterization techniques (TEM, SEM, DLS, zeta potential, BET, UV–vis). The results show no observable differences in their shapes and sizes (spherical and irregular, average size of 28 nm) while their surface charges changed in the cream, indicating no major modification of their primary sizes, morphology, and the corresponding functional properties. They were present as individually dispersed nanoparticles and as groups or clusters of physically separated primary nanoparticles in both dry form and cream medium, showing suitable stability. Examination of AuNPs in a cosmetic cream is challenging due to the required conditions of various characterization techniques but necessary for obtaining a clear understanding of the AuNPs’ properties in cosmetic products as the surrounding medium is a critical factor for determining their beneficial or harmful effects in cosmetic products.

## 1. Introduction

Gold nanoparticles (AuNPs) have now been used in luxury cosmetic products such as skin creams, lotions, hair products, facial masks, lipsticks, deodorants, etc. for several years, with declared beneficial anti-aging, hydrating, restorative, and regenerative properties [1,2,3,4,5]. AuNPs are considered as a valuable material in cosmeceutical products (cosmetic products with pharmaceutical, medically beneficial active ingredients) due to their strong antifungal and antibacterial properties. Nanomaterials in cosmeceuticals enhance dermal penetration and control the release of active ingredients, improve stability and moisturizing power, and also act as active ingredients themselves [3]. Well-known cosmetic brand companies, such as L’Oreal, L’Core Paris, Chantecaille, and Orogold use AuNPs to produce more effective creams and lotions [6]. The main beneficial effects of nanogold in beauty care studied by these manufacturers are acceleration of blood circulation, cell regeneration stimulation and skin damage repair, anti-inflammatory properties, antiseptic properties, improved skin firmness and elasticity, delayed aging process, collagen production promotion, and skin metabolism vitalization [5,7].

There are several studies on the safety and functionality of using nanomaterials and AuNPs as cosmetic ingredients [3,4,8,9]. A thorough knowledge of the harmful effects of these nanoparticles has not been established for the use of nanomaterials in cosmetics, while it is accepted that the sizes, shapes, concentrations, and frequency of use have a great impact on the effects of nanomaterials applied on the skin [2,10]. Before releasing a cosmetic product in the European Union (EU) market, manufacturers are required to report their ingredients according to the regulation on cosmetic products No. 1223/2009. For cosmetics with nanomaterial ingredients, testing information needs to be submitted for an evaluation of any potential adverse effects of the product carried out by the Scientific Committee on Consumer Safety (SCCS) of the EU. Several manufacturers have reported the use of AuNPs in various forms in their products, analyzing these ingredients with standard tests commonly used for cosmetics. However, these tests were not designed for nanoparticles, making the use of AuNPs and their potentially detrimental effects unknown. The lack of necessary information from producers registering their AuNPs containing cosmetics in the EU resulted in an inconclusive opinion on the use of these nanoparticles in cosmetics, with a safety concern for using them issued in a report by the SCCS of the EU [11]. Additionally, the SCCS issued inconclusive decisions regarding other metallic nanoparticles in cosmetics, such as silver, platinum, and copper, due to missing information on their behavior [12,13,14]. As the size, shape, surface charge, concentration, etc. affect their performance, these properties should be studied in a cosmetic medium. Such studies are not easily available, in contrast to the typical studies of raw nanoparticle materials.

As cosmetic creams are complex media with several ingredients, the characterization of primary nanoparticles in creams is challenging [10], with protocols developed for separating the nanoparticles from the product for analysis [1]. In practice, a cosmetic product is used in its entirety, not as single ingredients, and some arguments are made as to the applicability of safety studies of individual ingredients. As they are a single substance in a skin care product, other substances affect their physical and chemical characteristics, altering their behavior and making their toxicological aspects not easily translated to various cosmetic formulations [2].

A production method for AuNPs was developed using a technique called ultrasonic spray pyrolysis (USP) [15]. This method produces AuNPs in large quantities, in an aqueous suspension with stabilizers to reduce agglomeration, providing for longer-term stability of nanoparticles. Freeze-drying this suspension produces dried AuNPs in a stabilizer matrix cake, which increases the shelf life of the AuNPs, and can then also be used to disperse them in nonaqueous media. AuNPs produced by USP and freeze drying have been used successfully to produce cosmetic skin care creams, with initial testing of the AuNPs creams’ beneficial properties on volunteers [16].

As the beneficial or harmful effects of AuNPs in cosmetic creams are not conclusive, an investigation of the physical, chemical, and surface properties of AuNPs in the produced cosmetic skin care creams is conducted in this study. These properties determine the effects of AuNPs in cosmetic applications. An expectation of the behavior of the considered nanoparticles and their functional characteristics in the surrounding cream medium is given based on their characterization and available information on the safety of similar nanoparticles in cosmetic creams.

## 2. Materials and Methods

### 2.1. AuNPs Suspension Synthesis

The AuNPs were produced with a custom USP device from Zlatarna Celje d.o.o., Slovenia [17]. The device uses a specially designed aerosol generator with a 1.6 MHz piezoelectric transducer (Liquifog II, Johnson Matthey Piezo Products GmbH, Germany) for aerosolization of the initial Au precursor solution. The used Au precursor solution was a dissolved Au chloride (HAuCl_4_, Glentham Life Sciences, UK) in deionized water, with a concentration of 1.0 g Au/L (2.0 g HAuCl_4_/L). Aerosolized precursor solution droplets were transferred into a tube furnace with a nitrogen carrier gas at a flow of 6 L/min. The tube furnace with a 35 mm diameter quartz tube used three heating zones, each with a length of 40 cm. The heating zone temperatures were set at 200, 400, and 400 °C. The first heating zone was used for precursor droplet evaporation. After the first heating zone, the reaction gas, hydrogen, was introduced into the tube furnace at a gas flow of 5 L/min. The second and third heating zones were used for reactions of dried precursor particles with hydrogen and formation of AuNPs. The gas flow then carries the AuNPs into three consecutively connected gas washing bottles with a collection medium. The collection medium used was deionized water with an added 2.5 g polyvinylpyrrolidone/L (PVP; average molecular weight, 50,000 g/mol; stabilizing agent; Sigma Aldrich, Germany) for preventing the AuNPs’ agglomeration. The synthesis was performed for 12 h. Afterwards, the produced AuNPs suspension was concentrated further with a rotary evaporation device, Büchi Rotavapor R-300 (Büchi Labortechnik AG, Flawil, Switzerland). Vapor pressure was set at 25 mPa, with a 195 rpm rotation speed and bath temperature of 35 °C. The final volume of the obtained suspension was 6.3 L. ICP-MS (Agilent 7500 CE, Santa Clara, CA, USA) confirmed that the final AuNPs suspension’s gold concentration was 148.1 mg/L, yielding a total mass of 933.03 mg of AuNPs.

### 2.2. AuNPs’ Freeze Drying

A Labfreez Instruments FD-200F SERIES (Labfreez Instruments Group Co., Ltd., Beijing, China) was used for freeze-drying the AuNPs suspension produced by USP. Freeze drying was used previously to dry nanoparticle suspensions. Eighty mL of AuNPs suspensions were placed in several wide-neck crystallizing dishes in a freeze-drying chamber, with a total suspension volume of 4 liters. The initial freezing was set at ambient pressure with a shelf temperature of −40 °C for 4 h. After freezing, sublimation was induced at a pressure of 1–4 Pa and a shelf temperature of +20 °C for 12 h. Finally, drying was completed after 30 h at a pressure of 1–4 Pa and a shelf temperature of +30 °C. The final dried products were cakes of the AuNPs embedded in a PVP matrix. The freeze-drying parameters were selected based on several trial runs of AuNPs suspensions for drying optimization, with the aim of producing uncollapsed cakes economically and in a relatively short time.

### 2.3. Cosmetic Skin Care Cream Preparation with Dried AuNPs

The dried AuNPs were used in a standard procedure for the preparation of cosmetic creams, with a weight percentage of 12% dried AuNPs embedded in a PVP matrix, 48% aqua purificata, and 40% commercial cream base (“U”, Unichem Pharm). The creams were prepared using a procedure with an individually programmable mixing machine, Unguator E/S (Gako International). The filling jar size used for the cream was 500/600 mL (rated volume/filling volume). The skin care cream containing the AuNPs was analyzed with various characterization techniques in this study three months after production, over the course of several weeks.

### 2.4. Characterization of AuNP Physicochemical Properties Using Different Analytical Techniques

#### 2.4.1. Transmission Electron Microscopy (TEM)

For the TEM analysis, a JEOL 2100 (JEOL, Japan) and a JEOL JEM-2200FS HR (JEOL, Japan) were used for the dried AuNPs and the creams. The samples were first dispersed in ethanol and added dropwise onto a copper TEM grid with an amorphous carbon film. The prepared TEM grids were dried before being used in the TEM examinations.

#### 2.4.2. Scanning Electron Microscopy (SEM)

Scanning electron microscopy with an energy-dispersive X-ray spectroscope (SEM-EDX) was conducted on a Sirion 400 NC (FEI, Hillsboro, OR, USA) and a JEOL JSM-IT800SHL with EDX (JEOL, Akishima, Tokyo, Japan). Freeze-dried AuNPs in PVP matrix samples were applied on a graphite tape in a thin layer on an SEM stub holder and moved into the sample chamber for analysis. The cosmetic cream with AuNPs was spread thinly on a graphite tape placed on the SEM stub holder before analysis.

#### 2.4.3. AuNPs’ Morphology

Morphology was determined with the analysis of TEM and SEM images using ImageJ software. The diameter, circularity, aspect ratio, and roundness of the nanoparticles were determined using the software. A total of 600 nanoparticles were analyzed for the determination of physical shape and size from several images.

#### 2.4.4. Dynamic Light Scattering (DLS) and Zeta Potential Measurements

For DLS analysis and zeta potential measurements, a suspension was prepared with distilled water and dried AuNPs. The prepared concentration was 0.00172 g/mL (grams of NP per ml of distilled water). The suspensions had a pH level of 3.56. The sample was moved into an omega cuvette and put into a DLS analysis machine. The analysis was conducted on a Malvern Zetasizer Nano ZS instrument (Malvern Panalytical, Worcestershire, UK). The results are reported as a mean value from six measurements.

#### 2.4.5. Inductively Coupled Plasma-Mass Spectrometry (ICP-MS)

The Au concentration of the AuNPs suspension was measured using inductively coupled plasma-mass spectrometry (ICP-MS) with an HP Agilent 7500 CE equipped with a collision cell (Santa Clara, CA, USA). The ICP-MS power was 1.5 kW, with a Meinhard Nebuliser, plasma gas flow of 15 L/min, nebulizer gas flow of 0.85 L/min, make-up gas flow of 0.28 L/min, and reaction gas flow of 4.0 mL/min. The instrument was calibrated with matrix-matched calibration solutions. The relative measurement uncertainty was estimated as ±3%.

#### 2.4.6. Brunauer–Emmett–Teller (BET) Surface Area Analysis

BET analysis was used to determine the specific surface area and volume-specific surface area using a TriStar II 3020 V1.03 (V1.03). The sample of a dried AuNPs’ raw material weighed 0.1973 g.

#### 2.4.7. Ultraviolet–Visible Spectroscopy (UV–Vis)

UV–vis measurements were conducted with a suspension prepared with distilled water and dried AuNPs. The concentration used for the measurement was 0.00172 g/mL (grams of NP per ml of distilled water). Three hundred μL of the sample were transferred to a flat bottom transparent polystyrene plate and measured in a UV–vis spectrometer, Agilent Cary 60, in the range of 400–1000 nm. The results are reported as a mean value from six measurements.

## 3. Results

The TEM and SEM images of the dried AuNPs and AuNPs in cosmetic creams were analyzed to determine their morphology before and after embedment in cosmetic creams in relation to their properties (Figure 1). In the EDX analysis of the dried AuNPs samples, carbon and oxygen could be observed in addition to gold at the analyzed locations where AuNPs were present. Carbon was present because of the graphite tape used to prepare the samples. Oxygen, as well as carbon, was also present as the dried AuNPs samples contained PVP as the stabilizing agent. No other elements were detected as a result of contamination or impurities of the produced dried AuNPs using EDX.

The morphology study showed spherical and irregular shapes of the dried AuNPs. An example of morphology determination from the TEM images is shown in Figure 2, with the values for size, circularity, aspect ratio, and roundness given in Table 1. The average size of the dried AuNPs was 28 ± 20 nm, determined from measuring 600 nanoparticles. A size distribution chart is also given in Figure 2. The chart shows that 95% of the AuNPs were in the size range below 75 nm. Some larger nanoparticles were also detected, up to a few hundred nm, along with bigger AuNPs clusters with sizes of more than a couple of hundred nanometers. A closer examination of these showed groups of smaller nanoparticles forming soft agglomerates that might break up when the dried nanoparticles are redispersed in a suspension and sonicated.

The determined average circularity was 0.931 for the measured AuNPs. The range of circularity was 0–1, where 1 represents a perfect circle with a smooth edge. As the value approaches 0, it indicates an increasingly elongated shape, or a shape with increasingly rough and sharp edges. The mean aspect ratio was determined to be 1.186, showing that most nanoparticles had similar diameters in different directions, which were measured as the major and minor axes. The mean roundness was at 0.852, showing the nanoparticle shapes as relatively close to being round, as this parameter is related more closely to the aspect ratio while omitting the smoothness of the edges. It may be considered as the inverse of the aspect ratio, where a value of 1 denotes a completely round nanoparticle within a range of 0–1.

The SEM and TEM observations of the AuNPs embedded in the cosmetic cream showed relatively well-dispersed nanoparticles across the base substance of the cream, with individual AuNPs and some smaller clusters visible in the studied samples. The clusters appeared not to be agglomerated, confirming the observation of soft agglomerates in the dried AuNPs cake. The AuNPs were well-integrated in the cream, making it difficult for the EDX examination to confirm the presence of Au in the investigated nanoparticles. The analysis of the samples was quite difficult, as the cosmetic cream base is organic, and samples become contaminated very quickly. The EDX signal of the AuNPs is also low, but visible. This effect was especially dominant in the TEM examinations. The AuNPs in the cosmetic cream were observed directly without any extraction processes which could change the AuNPs’ characteristics.

The DLS measurements were conducted on AuNPs suspensions with a prepared concentration of 0.00172 g/mL (grams of dried AuNPs per ml of distilled water). The obtained hydrodynamic diameter and size distribution are shown in Figure 3 and Table 2 and Table 3. The measured sample had a distinct peak in the size range of 100–700 nm and an additional peak in the size range of 20–60 nm. The measured hydrodynamic diameter was 303.7 nm, with the polydispersion index at 27.98%. The zeta potential distribution is shown in Figure 3, with a measured average zeta potential of −4.51 mV (Table 4).

The specific surface area (SSA) and volume-specific surface area (VSSA) determined using BET are shown in Table 5. The measured samples were the dried AuNPs in a PVP matrix with a weight of 0.1973 g.

The UV–vis measurements are shown in Figure 4, with a visible UV absorption peak for the AuNPs at 532 nm.

## 4. Discussion

The characterized dried AuNPs were compared to the AuNPs from the cosmetic cream for any changes that could be detected in their structure and morphology. The Au content of the nanoparticles was confirmed, even with difficulties regarding obtaining a clear EDX signal in SEM and TEM with the cosmetic cream’s AuNPs samples. The other elements, such as carbon, oxygen, sodium, sulfur, silicon, etc., were present in the medium surrounding the AuNPs in dried form as a stabilizer or in the cream base (see Figure 1 and Figure 2). No other impurities were detected. Extracting nanoparticles from skin creams was shown to completely alter their surface properties, which affects their hydrodynamic sizes [2] and may thus also affect their morphological properties due to agglomeration or other interactions with the cosmetic ingredients during the extraction procedure. Depending on the type of nanomaterial or cosmetic ingredients, the nanoparticles may have varied behaviors. An example is silver nanoparticles agglomerating in some cosmetic creams, while AuNPs did not agglomerate in the same medium [10]. Silver nanoparticles are known to have strong antibacterial properties, making them useful as a disinfectant in a number of cosmetic uses, such as underarm deodorants, or in toothpaste [18].

Comparison of the morphology of the dried AuNPs and the AuNPs from the cosmetic cream did not show changes in the nanoparticles’ structure or shapes with regard to agglomeration and clustering. The sizes and shapes were similar in both aspects, with no major changes detected when the AuNPs were integrated in the cosmetic cream. The circularity and aspect ratio showed that the nanoparticles ranged from spherical to somewhat irregular shapes, as seen in the SEM and TEM images. The selected parameters with the USP and the freeze-drying method did not produce perfectly spherical AuNPs, while the size distribution of the nanoparticles was relatively narrow as most of the particles were <37.5 nm (83.54%), together with some additional mid-sized nanoparticles of up to 75 nm (total 95.74%) and a smaller number of larger particles of up to 200 nm (the maximum size of the measured particles in the sample was 204 nm).

The nanoparticle distribution across the samples showed some monodispersed nanoparticles, along with groups or clusters of smaller nanoparticles that seemed to be separated physically one from the other, yet still in the same group’s vicinity. There was a higher number of these groups detected in the cosmetic cream; however, this may be due to the higher difficulty of detecting singular nanoparticles in the cream medium. As these groups are also apparent in the dried AuNPs samples, the formation of such nanoparticle arrangements may be attributed to the stabilizing agent, PVP. The tendency to form these groups is retained with mixing the dried AuNPs in the cream base. This may also be the case with redispersing the dried AuNPs into a suspension as the DLS measurements showed a larger hydrodynamic nanoparticle size than that observed with measuring the primary nanoparticles with SEM or TEM.

The polydispersion index in DLS measurements is a dimensionless measure of the broadness of size distribution and lies between 0% and 100%, with zero for monodisperse nanoparticles [19]. In our case, the polydispersion index was relatively low; therefore, we can assume a semimonodispersed suspension. Only two distinct peaks were observed, as seen in Table 3. The large size distributions and hydrodynamic diameters can be attributed to soft agglomerates and nanoparticle clustering as the SEM and TEM showed a relatively small number of larger nanoparticles. The average zeta potential of −4.51 mV shows a low electrostatic stability for AuNPs (Table 4). Steric stabilization of AuNPs with PVP is, therefore, the dominant stabilization mechanism [20,21]. The observed behavior of the produced dried AuNPs when mixed with water may also contribute to understanding their characteristics after mixing them in cosmetic creams.

The AuNPs suspension kept its red color when exposed to visible light, as evidenced by the UV–vis measurements, which suggests the preservation of nanoparticle size after the redispersion of the dried AuNPs. The absorption peak of 532 nm was similar to the maximum absorption peaks at 520–530 nm for the AuNPs with sizes between 12 and 41 nm [22]. The small shift of the absorption spectra was caused by the size distribution of the AuNPs, although the distinct absorption peak observed shows that higher rates of agglomeration did not occur as nanoparticles of larger, several hundred nanometer sizes, do not preserve the surface plasmon effect required for increased absorption at this light wavelength. As the PVP stabilizer preserves the primary nanoparticles from agglomeration and their surface plasmon resonance, the DLS results imply the grouping of nanoparticles and interference with the measurements.

The presence of the PVP stabilizer is critical for AuNPs’ functional properties as removing PVP results in an increased effect of agglomeration. The BET measurements were conducted on the PVP–AuNPs cake, and the VSSA results were less than 60 m^2^/cm^3^, which was defined as the value obtained for nanoparticles using the European Commission guidelines [23]. The measurement of the specific surface area of materials using the BET method works on the principle of physisorption of nitrogen (or other gas) on the surface of a vacuumed sample. The isothermal pressure change curve of absorption allows the specific surface area of the sample to be calculated. The method does not allow selective measurement of the specific surface area of individual phases or materials. From this point of view, the obtained results of the specific surface area of the porous PVP sample with embedded AuNPs do not represent the final surface area of the nanoparticles, but the entire specific surface area of the PVP matrix with AuNPs. The use of PVP is important for preventing agglomeration during USP synthesis and freeze drying as PVP retains the functional and morphological properties of the nanoparticles that are critical to their use. PVP as a component dissolves in polar or nonpolar solvents at end use and does not represent a harmful component in cosmetic products such as skin care creams and sprays.

This indicates that the surface area measured using the BET method is provided mostly by the PVP matrix. Removing PVP from the nanoparticles is likely to provide higher VSSA values, however, the surface plasmon resonance and other properties required for applications including cosmetic creams would be lost. A comparison of the dried AuNPs with or without a stabilizer is given in Figure 5, which shows an additional sample of AuNPs produced with the same parameters without the PVP stabilizer for stability studies in previous research. Formation of hard agglomerates removes the useful properties of AuNPs, so a stabilizer needs to be added for their use and in all the characterization methods. Hard agglomerates also cannot be redispersed into a suspension and remain as visible small black nanoparticles, sedimenting in an aqueous medium. In this case, primary nanoparticle measurements and establishing the degree of agglomeration from SEM and TEM are more relevant for determining the AuNPs’ size and shape properties, which are critical parameters in terms of their effects in cosmetic creams. The included PVP stabilizer is a common ingredient in cosmetic creams [24,25] and in pharmaceutics [26], so it is proven to be safe for use in cosmetics and personal care products.

An important factor for determining the effects of AuNPs in cosmetic creams is the permeation of these nanoparticles through the skin and their potential benefits or harmfulness as they pass through skin layers. These abilities are governed mainly by the shape, size, surface charge, and dose of these nanoparticles.

Very small nanoparticles < 6 nm show a high permeation, with the highest diffusivity observed for 2 nm AuNPs [8,27]. The size dependence on penetration was confirmed in rat skin, where 15–22 nm-sized AuNPs showed a higher diffusivity than 102–105 nm- and 186–198 nm-sized nanoparticles [28,29]. A study involving 28, 49, and 73 nm spherical AuNPs in use on the skin of mice also showed a size dependence for the nanoparticle skin penetration. This study also compared the penetration of raw AuNPs and AuNPs embedded in skin creams. The results showed some accumulation of AuNPs in the skin, but no accumulation was detected in the major organs, meaning that AuNPs cannot enter the bloodstream and be transported to the major organs [2]. Regarding surface charge as a factor for permeability, neutrally charged AuNPs had the highest penetration in the skin’s bilayer interior, while both positively and negatively charged AuNPs did not get into the bilayer [8].

Analysis of the first skin layer (stratum corneum) in contact with cosmetic creams showed that penetration through aqueous pores could take place for nanoparticles < 36 nm, while there is a higher possibility of penetration through the intercellular lipidic matrix for nanoparticles smaller than 5–7 nm. The larger space provided in the vicinity of hair follicles also allows for the accumulation of nanoparticles in their subsequent discharge from this area by sweat and sebum [3].

Other studies also suggest that metallic nanoparticles sized 4–20 nm can probably penetrate both intact and damaged skin, and nanoparticles sized 21–45 nm can penetrate only damaged skin, whereas nanoparticles > 45 nm cannot penetrate intact or damaged skin [30]. Physical characteristics also affect the nanoparticles’ accumulation and penetration pathways, with nanoparticle size being an important factor for penetration. In a study of skin penetration via hair follicles, it was shown that spherical nanoparticles had a relatively poor penetration compared to nanorods, while anisotropic particle shapes also contributed to particle accumulation [31].

Concerning the harmful effects after penetration, raw AuNPs with sizes of about 28 nm were shown to have some weak toxicological responses on hairless mice skin. AuNPs of larger sizes (49 and 73 nm) showed no toxicological response [2].

There are several more studies regarding the penetration and toxicity of AuNPs. However, the short overview of their morphology and surface properties on harmful effects in cosmetic creams shows that these effects are more prevalent as the nanoparticle size decreases. In our case, the AuNPs’ sizes and morphologies showed properties which suggest low harmfulness and penetration of the nanoparticles from the cosmetic cream into the lower skin layers. The sizes are somewhat above the threshold for deeper penetration while the morphology (spherical and irregular shapes) and measured surface charge showed an average tendency for penetration and accumulation.

The present study showed that when these nanoparticles are embedded in cosmetic creams, their primary sizes, morphologies, and optical properties remain intact. A preliminary study of these cosmetic creams on volunteers showed no detrimental effects on the skin [16] and a high probability that the embedding of these AuNPs in cosmetic creams reduces their toxicological response if such an effect is present in raw AuNPs. Based on another study, the interaction between AuNPs and the ingredients of cosmetics seems to play a critical role in toxicity enhancement or reduction [2].

As the use of these materials with cosmetic ingredients alters their interactions, they should also be tested in relevant cosmetic formulations. This impacts the characterization methods as it is difficult to obtain relevant information on AuNPs’ characteristics while the tests for harmful effects, such as acute toxicity, irritation, sensitization, mutagenicity, genotoxicity, reproductive toxicity, carcinogenicity, etc., should remain relevant. As nanoparticles are a relatively novel substance in cosmetics, the information on them is a major lacking point, as highlighted by the SCCS of the EU Commission [11]. As the existing testing methods are being used to obtain these data, newly developed methods for nanomaterials should thus include the entire cosmetic products containing AuNPs, without their isolation for testing purposes.

### Study Limitations

The study on the properties of AuNPs was conducted on skin creams produced with a custom nanoparticle synthesis method, with specifics regarding the obtained particle size, shape, and surface charge. The skin cream formulation also used proprietary ingredients of the commercial cream base complex. This resulted in the observed behavior of the studied AuNPs, which may vary with other production methods, cosmetic formulations, and integration of nanoparticles into creams at elevated temperatures, during vigorous stirring, or with other techniques that may affect their interaction with the cream complex and their stability. As such, the characteristics of the AuNPs described in this study may not be easily translated to other cosmetics containing nanoparticles, for which separate studies are needed.

Additionally, as s cosmetic product ages, degradation of its ingredients in the cream base again alters the surface charge of the AuNPs, and the stability, degree of agglomeration, and morphology may not be assumed to be the same as with a fresh formulation. Dry AuNPs in a PVP matrix may be stable up to several years if stored properly away from moisture as compared to several months for AuNPs in a colloid suspension. It may be assumed that for AuNPs in creams, similarly to colloid AuNPs suspensions, aging and exposure to elevated temperatures or direct sunlight would negatively affect their stability, as the cream ingredients would also deteriorate. However, further studies of aging AuNP creams would be reasonable to determine nanoparticle behavior, such as over the course of an average cream usage timespan.

## 5. Conclusions

The following conclusions can be drawn from the characterization study of AuNPs embedded in cosmetic skin care creams:AuNPs with a PVP stabilizer are present as individually dispersed nanoparticles and as groups of physically separated primary nanoparticles, which are not hard agglomerates that cannot be split up. These groups of AuNPs are present in the freeze-dried form, as well as in the cosmetic cream.Extraction of AuNPs from cosmetic creams may alter their characteristics, making it necessary to characterize them in the creams, although it is difficult to determine their morphology and surface properties when inside the cosmetic cream medium.The stability of AuNPs in creams is not affected when they are inserted into the cream medium as their primary sizes, morphologies, and optical properties remain comparable.The tests for beneficial or harmful effects of AuNPs should include the entire cosmetic products containing these nanoparticles, without their isolation for testing purposes.Further studies are recommended to determine the characteristics of AuNPs in aging skin creams over a typical cream usage timespan.

## Figures and Tables

**Figure 1 materials-16-03011-f001:**
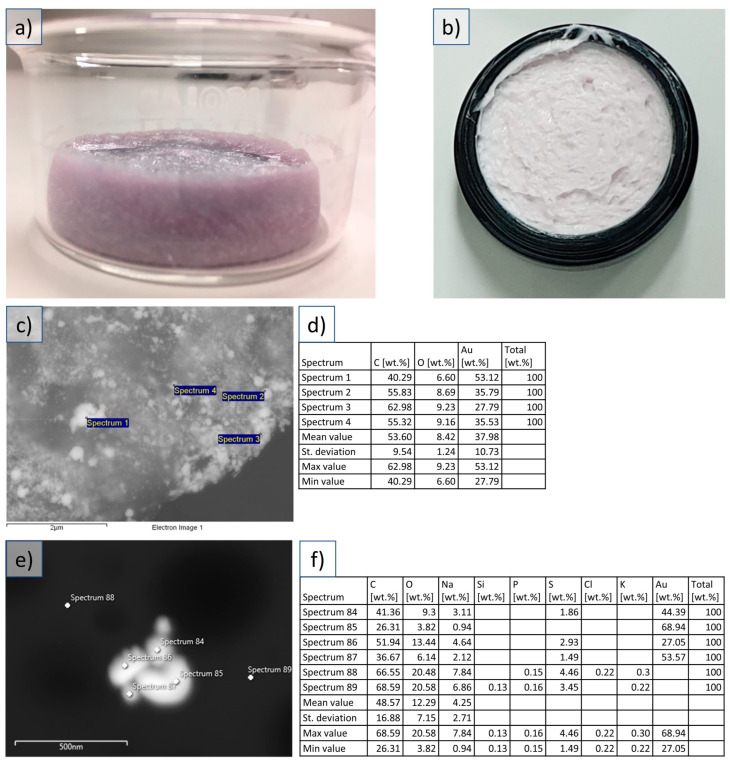
Dried AuNPs, cosmetic cream with AuNPs, and SEM examinations. (**a**) Freeze-dried PVP cake with AuNPs. (**b**) Cosmetic cream with embedded AuNPs. (**c**) SEM image of the dried AuNPs. (**d**) EDX analysis of select points in the corresponding SEM image of the dried AuNPs. (**e**) SEM image of the AuNPs embedded in the cosmetic cream. (**f**) EDX analysis of select points in the corresponding SEM image of the AuNPs embedded in the cosmetic cream.

**Figure 2 materials-16-03011-f002:**
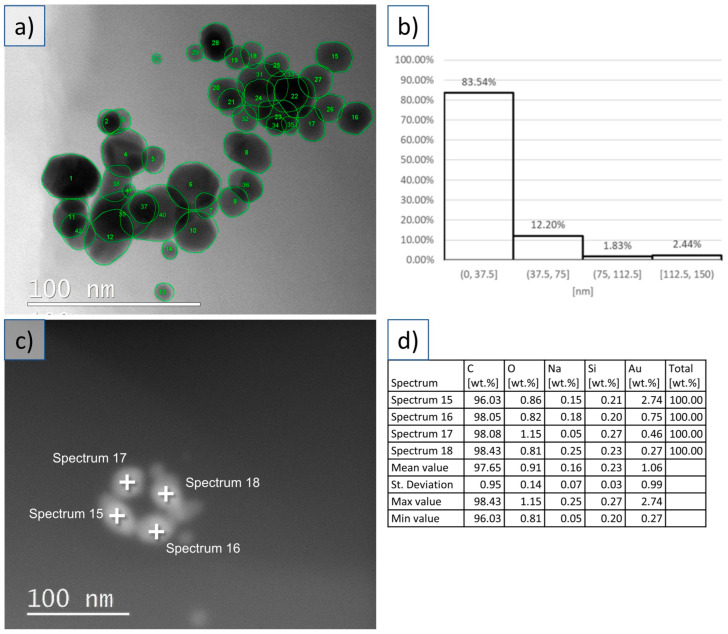
TEM investigations of the dried AuNPs and the cosmetic cream with AuNPs. (**a**) TEM image of the dried AuNPs with particle size and morphology analysis. (**b**) Size distribution of the analyzed AuNPs. (**c**) TEM image of the AuNPs embedded in the cosmetic cream. (**d**) EDX analysis of select points in the corresponding TEM image of the AuNPs embedded in the cosmetic cream.

**Figure 3 materials-16-03011-f003:**
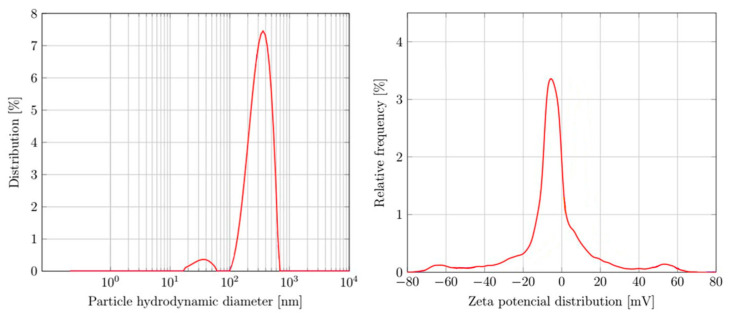
Particle size distribution from the DLS measurements and zeta potential distribution.

**Figure 4 materials-16-03011-f004:**
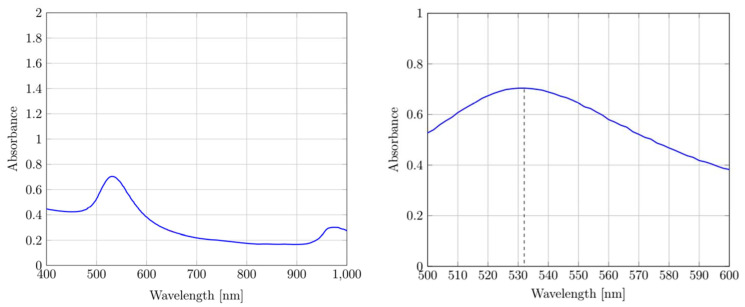
UV–vis measurement spectrum with a signature peak for absorption.

**Figure 5 materials-16-03011-f005:**
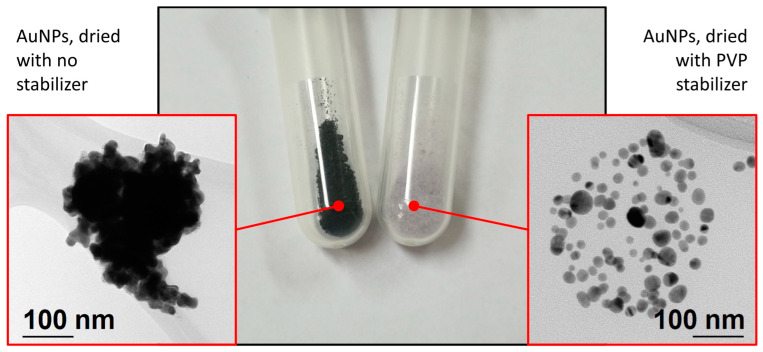
Comparison of the dried AuNPs’ dispersion and agglomeration from TEM images, with and without a stabilizer.

**Table 1 materials-16-03011-t001:** Obtained AuNPs’ size and morphology parameters: circularity, aspect ratio, and roundness.

	Particle Diameter (nm)	Circularity	Aspect Ratio	Roundness
Mean value	28.03	0.931	1.186	0.852
Standard deviation	19.76	0.024	0.129	0.085
Maximum value	204.55	0.978	1.587	0.990
Minimum value	10.60	0.830	1.010	0.630

**Table 2 materials-16-03011-t002:** DLS measurements’ average particle size result.

Hydrodynamic Diameter (nm)	Polydispersion Index (%)
303.7	27.98

**Table 3 materials-16-03011-t003:** Particle size distribution from the DLS measurements.

Peak	Size (nm)	Intensity (%)	Standard Deviation (nm)
1	429.30	87.39	121.20
2	26.47	12.61	5.31
3	0	0	0

**Table 4 materials-16-03011-t004:** Zeta potential measurements results.

Average Zeta Potential (mV)	Standard Deviation (mV)	pH
−4.51	0.47	3.56

**Table 5 materials-16-03011-t005:** SSA and VSSA of the dried AuNPs in a PVP matrix.

SSA-BET (m^2^/g)	Au Density (g/cm^3^)	VSSA (m^2^/cm^3^)
1.448	19.3	27.959

## Data Availability

The data presented in this study are available upon request from the corresponding author.

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
