# Peer review of "Physicochemical Properties of Gold Nanoparticles for Skin Care Creams"

_materials, 2023, doi:10.3390/ma16083011_

Round 1

Reviewer 1 Report

This manuscript reports an interesting study of gold nanoparticles in skin care creams. The authors synthesized gold nanoparticles by thermally reducing 

1. The title should mention the specific properties of gold nanoparticles studied in this work instead of using the general term of 'properties'.

2. The particle size analysis in Figure 2 seems very subjective. Why some of the particles are not selected for analysis? Details on the method are needed.

3. What is the yield of the Au NP synthesis?

Author Response

This manuscript reports an interesting study of gold nanoparticles in skin care creams. The authors synthesized gold nanoparticles by thermally reducing 

  1. The title should mention the specific properties of gold nanoparticles studied in this work instead of using the general term of 'properties'.

The title was changed to »Gold nanoparticles sizes, shapes and stability in skin care creams«.

  1. The particle size analysis in Figure 2 seems very subjective. Why some of the particles are not selected for analysis? Details on the method are needed.

The image was selected only as a visual example of measuring the particles during the performing of this task with the ImageJ software, as described in chapter 2.4.3. AuNPs' morphology. All of the visible particles with distinguishable edges were actually selected for analysis. Figure 2 a) is now replaced with another image of a higher magnification, showing the complete analysis of the TEM image, with better visibility of the individual particles with their shapes and sizes.

  1. What is the yield of the Au NP synthesis?

The final volume of the obtained suspension was 6.3 litres, with a concentration of 148.1 mg/L, which yielded a total mass of 933.03 mg of AuNPs (about 1 g). The yield was added in chapter 2.1. AuNPs` suspension synthesis.

Reviewer 2 Report

COMMENTS FOR AUTHORS 

The authors presented a research on the difference in produced dried AuNPs and AuNPs in a cosmetic cream for their size, morphology and surface changes when embedded in the cream. The manuscript is reasonable in structure and detailed in content. However, some problems still exist. It is advised to accept them after modification.

1. Table 3 is not mentioned in the manuscript.

2. The authors should explain the abbreviations of the words in the manuscript, such as “EU” in line 41 and “SCCS” in line 42.

3. The delimiters between keywords should be unified.

4. The authors could consider modify the title of the section 2.4 “AuNPs` characterisation”.

5. In the line 387, the authors mention “The extraction of AuNPs from cosmetic creams may alter their characteristics”, more detailed demonstration is needed in the content.

Author Response

Review 2

The authors presented a research on the difference in produced dried AuNPs and AuNPs in a cosmetic cream for their size, morphology and surface changes when embedded in the cream. The manuscript is reasonable in structure and detailed in content. However, some problems still exist. It is advised to accept them after modification.

  1. Table 3 is not mentioned in the manuscript.

Added reference to Table 3 in line 217 on page 7.

  1. The authors should explain the abbreviations of the words in the manuscript, such as “EU” in line 41 and “SCCS” in line 42.

Added abbreviation explanations at their first appearance in the text on page 1, European Union (EU), Scientific Committee on Consumer Safety (SCCS).

  1. The delimiters between keywords should be unified.

Corrected to semicolon between keywords.

  1. The authors could consider modify the title of the section 2.4 “AuNPs` characterisation”.

The section title was changed to “Characterisation of AuNP physicochemical properties by different analytical techniques«

  1. In the line 387, the authors mention “The extraction of AuNPs from cosmetic creams may alter their characteristics”, more detailed demonstration is needed in the content.

Bearing in mind that extracted AuNPs from cosmetic creams can change their physico-chemical surface characteristics because they have interacted with other cream ingredients, it is necessary to characterize them in creams. With this request, we encounter the difficulty of determining their morphology and surface. Additional clarification with references was added in the discussion section in lines 277-283:

Extracting nanoparticles from skin creams was shown to completely alter their surface properties, which affects their hydrodynamic sizes [2], and may thus also affect their morphological properties due to agglomeration or other interactions during the extraction procedure. Depending on the type of nanomaterial or cosmetic ingredients, the nanoparti-cles may have varied behaviours. An example is silver nanoparticles agglomerating in some cosmetic creams, while AuNPs did not agglomerate in the same medium [10].

Reviewer 3 Report

The paper presented for review is an interesting study on the qualitative assessment of gold nanoparticles in the presence of a factor protecting against aggregation and in a cosmetic base.

This technological paper illustrates how gold nanoparticles behave in a protected form and under conditions mimicking the conditions in which they are actually intended to be used.

The abstract needs improvement, especially the first sentence, which contains a logical error. I also ask for a structured breakdown and an indication of the scientific background,  research methods, the most important results and conclusions.

The manuscript begins with a short introduction. I am asking for a broader discussion of the purpose for which gold nanoparticles are used in cosmetics. Please present what research has already been carried out on this subject and what the legal provisions regarding this raw material look like.

Material and method: please indicate whether the research was performed once or in repetitions, and if so, how many repetitions were performed for a given observation.

Results: This chapter will require the most revisions. Here, only the results with objectivity should be presented. It is not considered appropriate to cite other authors in the results section. Discussion of the results and their confrontation with the results of other authors are issues that should appear in the chapter: Discussion. I would like to ask you to remove extensive discussions from the chapter: Results, and moving to the next chapter.

Discussion: please indicate in this chapter whether similar studies have ever been conducted for silver, copper or platinum nanoparticles. These metals in the form of nanoparticles are also used in the cosmetics industry, which is why it seems interesting to compare the technological parameters for these raw materials.

Please, add one more subchapter to the manuscript: study limitation. In this subchapter, I would like to ask you to indicate the limitations of this work and the conclusions that have been drawn. Please pay special attention to the time that elapsed from the preparation of the cream to its examination. Will an aging cosmetic change its properties and the behavior of gold nanoparticles in such a cosmetic may change?

I also ask you to consistently use the impersonal form of expression.

Author Response

REVIEW 3:

The paper presented for review is an interesting study on the qualitative assessment of gold nanoparticles in the presence of a factor protecting against aggregation and in a cosmetic base.

This technological paper illustrates how gold nanoparticles behave in a protected form and under conditions mimicking the conditions in which they are actually intended to be used.

The abstract needs improvement, especially the first sentence, which contains a logical error. I also ask for a structured breakdown and an indication of the scientific background,  research methods, the most important results and conclusions.

The abstract was revised for greater clarity, taking the suggested comments into consideration.

The manuscript begins with a short introduction. I am asking for a broader discussion of the purpose for which gold nanoparticles are used in cosmetics. Please present what research has already been carried out on this subject and what the legal provisions regarding this raw material look like.

The discussion with the use of gold nanoparticles, their beneficial effects and the legal provisions has been expanded in the introduction, with some references added.

Material and method: please indicate whether the research was performed once or in repetitions, and if so, how many repetitions were performed for a given observation.

The AuNP synthesis and freeze-drying was based on previous productions, the parameters were selected based on the optimal conditions for synthesizing these nanoparticles and for their inclusion in skin cream. The dry AuNPs and creams with AuNPs were used as the samples for the different analysis techniques. Upon succesful analysis, such as for example TEM or SEM, no other iterations were deemed necessary. Additionally, the number of measurements was added to the different techniques, where the samples were analysed in repetitions.

Results: This chapter will require the most revisions. Here, only the results with objectivity should be presented. It is not considered appropriate to cite other authors in the results section. Discussion of the results and their confrontation with the results of other authors are issues that should appear in the chapter: Discussion. I would like to ask you to remove extensive discussions from the chapter: Results, and moving to the next chapter.

The discussion of results has been moved into the discussion chapter from the results chapter, where only the details from the measurements are now presented.

Discussion: please indicate in this chapter whether similar studies have ever been conducted for silver, copper or platinum nanoparticles. These metals in the form of nanoparticles are also used in the cosmetics industry, which is why it seems interesting to compare the technological parameters for these raw materials.

Some examples were given for comparison with other metallic nanoparticles in the discussion, such as silver nanoparticles agglomerating in a cosmetic medium, where gold nanoparticles did not agglomerate. In the introduction, mentions of SCCS opinions on silver, copper and platinum nanoparticles were included, which are also inconclusive due to missing information. Similar studies for these nanoparticles in the cosmetic medium are not easily available. Some comparisons were added, however, the use of the many other nanoparticles in various cosmetics presents a broad topic, and we believe that a more comprehensive comparison (for example in the discussion section) would deviate greatly from the presented results and is therefore more suitable for a review article.

Please, add one more subchapter to the manuscript: study limitation. In this subchapter, I would like to ask you to indicate the limitations of this work and the conclusions that have been drawn. Please pay special attention to the time that elapsed from the preparation of the cream to its examination. Will an aging cosmetic change its properties and the behavior of gold nanoparticles in such a cosmetic may change?

A subchapter for study limitation was added to the discussion section. The subchapter text considers different cosmetic formulations, AuNP production techniques, and cosmetic ageing with degradation of the ingredients. The age of the studied cream was added in the materials and methods section, in the subchapter for preparation of the cream with AuNPs.

I also ask you to consistently use the impersonal form of expression.

The text was checked for the use of passive voice.
